# An Overview of the Practices and Management Methods for Enhancing Seed Production in Conifer Plantations for Commercial Use

**Yan Li [1,2], Xiang Li [1], Ming-Hui Zhao [1], Zhong-Yi Pang [3], Jia-Tong Wei [1], Mulualem Tigabu [4], Vincent L. Chiang [1,5], Heike Sederoff [5], Ronald Sederoff [5] and Xi-Yang Zhao [1,2,*]**

1   State Key Laboratory of Tree Genetics and Breeding, School of Forestry, Northeast Forestry University, Harbin 150040, China; Ly2019nefu@163.com (Y.L.); lx2016bjfu@163.com (X.L.); zhaominghui66@163.com (M.-H.Z.); wjtnefu@163.com (J.-T.W.); vchiang@ncsu.edu (V.L.C.)
2   College of Forestry and Grassland, Jilin Agricultural University, Changchun 130118, China
3   State-Owned Xinmin Machinery Forest Farm, Shenyang 110300, China; 13478192389@163.com
4   Southern Swedish Forest Research Centre, Swedish University of Agricultural Sciences, SE-230 53 Alnarp, Sweden; mulualem.tigabu@slu.se
5   Forest Biotechnology Group, Department of Forestry and Environmental Resources, North Carolina State University, Raleigh, NC 27659, USA; heike.sederoff@yahoo.com (H.S.); ron_sederoff@ncsu.edu (R.S.)
*   Correspondence: zhaoxy@nefu.edu.cn; Tel.: +86-0451-8219-2225

**Abstract:** Flowering, the beginning of the reproductive growth, is a significant stage in the growth and development of plants. Conifers are economically and ecologically important, characterized by straight trunks and a good wood quality and, thus, conifer plantations are widely distributed around the world. In addition, conifer species have a good tolerance to biotic and abiotic stress, and a stronger survival ability. Seeds of some conifer species, such as *Pinus koraiensis*, are rich in vitamins, amino acids, mineral elements and other nutrients, which are used for food and medicine. Although conifers are the largest (giant sequoia) and oldest living plants (bristlecone pine), their growth cycle is relatively long, and the seed yield is unstable. In the present work, we reviewed selected literature and provide a comprehensive overview on the most influential factors and on the methods and techniques that can be adopted in order to improve flowering and seed production in conifers species. The review revealed that flowering and seed yields in conifers are affected by a variety of factors, such as pollen, temperature, light, water availability, nutrients, etc., and a number of management techniques, including topping off, pruning, fertilization, hormone treatment, supplementary pollination, etc. has been developed for improving cone yields. Furthermore, several flowering-related genes (*FT*, Flowering locus T and MADS-box, *MCMI*, *AGAMOUS*, *DEFICIENCES* and *SRF*) that play a crucial role in flowering in coniferous trees were identified. The results of this study can be useful for forest managers and for enhancing seed yields in conifer plantations for commercial use.

**Keywords:** conifers; flowering; seed production; pollination; phytohormones; tree management; nutrient fertilization

## 1. Introduction

Conifer species are typically tall perennial and evergreen trees or shrubs. They are the largest and most important species of gymnosperms with high economic and ecological values, which have a potential life span of 1000 years under natural growth conditions [1]. Conifers include about 613 species, mainly in the Pinaceae, Taxodiaceae and Cupressaceae families that are widely distributed over the world [2,3]. They appeared on earth's surface about three hundred million years ago [4]. The number of species of angiosperms is estimated at about 300,000, conifers dominated the forests of the Jurassic until the rise

of the angiosperms during the Cretaceous period [4]. Conifers also include the Araucaria family, the Podocarpus family, the species *Sciadopitys verticillata* and the yew family (Taxacae). Although conifers dominate many of the temperate forests of the Northern hemisphere, 208 (34%) of the 613 extant conifer species are threatened with extinction. Conifer species are also a crucial landscape tree species due to its green leaves and straight stems, and the world's five largest courtyard species are conifer species, such as *Cedrus deodara* (*Deodara cedar*), *Sequoiadendron giganteum* (giant sequoia), *Araucaria cunninghamii* (hoop pine) and *Pseudolarix amabilis* (golden larch) [5]. In addition, conifers supply over 50% of the world's timber, and most conifer wood is processed for pulp and paper [6]. *Larix* (Larch sps), *Picea asperata* (Chinese spruce) and *Platycladus orientalis* (Oriental arborvitae) have gradually become great timber tree species due to a good wood quality suitable for wood processing, papermaking and construction industries [7,8]. Seeds of some conifer species, such as *P. koraiensis*, have a high oil content and a variety of potential medicinal components, including vitamins, fatty acids and minerals, which are widely used in food industries and drug development with potential industrial applications and economic benefits [9,10].

Tree genetic improvement is an important technology for breeding new and improved varieties [11]. Reproduction through tissue culture and stem cuttings of many conifer species remains difficult; thus, direct sowing and planting of seedlings raised in nurseries is still considered a useful method for breeding an improved variety. Many seed orchards of conifer species have already been established [12,13], such as *P. koraiensis*, *Pinus sylvestris* and *Larix*, are in the primary or second-generation seed orchards [14–16]. While some tree species, such as *Pinus taeda* (loblolly pine) and *Pinus radiata* (Monterey pine), are in the fourth-generation of selection in seed orchards [17,18]. Most seed orchards show periodic and variable reproduction due to unfavorable environment conditions and challenges of management, which seriously affect the yield of seed orchards [19].

In this review, we discuss the crucial factors related to the flowering and seed production of conifer species. In addition, we synthesized the techniques and management practices that can be adopted to enhance seed production. In view of the complex challenges in defining the mechanisms that control flowering and reproduction, we propose some solutions to increase the economic benefits accrued from forestry activities, while providing a theoretical basis for genetic improvement and expanded planting of conifer species.

## 2. Materials and Methods

For the current study, we reviewed articles indexed in the databases Web of Science and Google Scholar. At the first stage, relevant studies were identified by using combinations of the following keywords: "conifer", "flower development", "seed production", "pollination", "phytohormones", "tree management", "topping off", "dwarf", "water and fertilizer coupling", "temperature", "light", "genetic improvement" and "seed orchards". At the final stage, the most appropriate articles were selected to perform a solid overview of (a) the factors affecting conifers' flowering and seed production and (b) the techniques and management practices that can be adopted to enhance seed production.

## 3. Results

The growth cycle of conifer species is relatively long, and many factors affect seed production, resulting in unstable seed yields. Thus, it is crucial to propose technical guidelines for improving seed production in conifer plantations for commercial use (Figure 1).

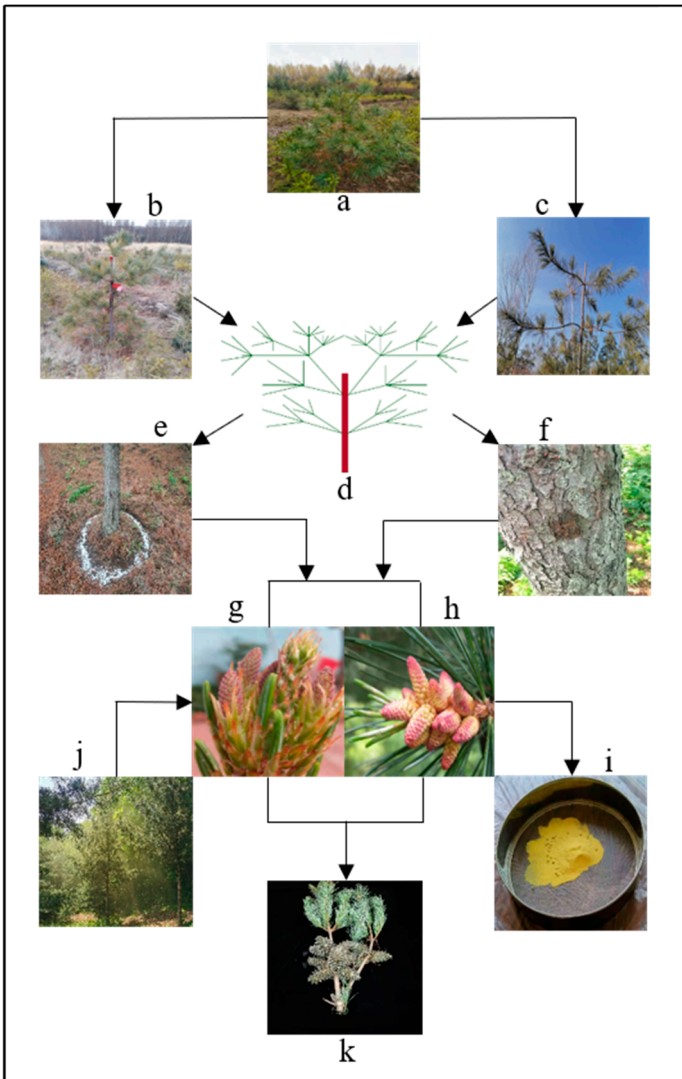

**Figure 1.** Flowering and seed yields in conifers are affected by a variety of factors (pollen, temperature, light, water availability, nutrients, etc.) and that for improving cone yields a number of management techniques have been tested, including topping off, pruning, fertilization, hormone regulation, supplementary pollination, etc. Panel (**a**) represents the 5–10-year-old tree of conifer species (*P. koraiensis* for example), panel (**b**) is topping off, panel (**c**) is pruning, panel (**d**) shows ideal tree shape, panel (**e**) shows fertilization and watering, panel (**f**) depicts hormone application, panels (**g**) and (**h**) are female and male cones, respectively, panel (**i**) shows pollen storage, panel (**j**) depicts supplementary pollination and panel (**k**) shows improved cone yield.

### 3.1. Factors Affecting Seed Formation and Development

#### 3.1.1. Pollen

Pollen, the material basis for transmitting genetic information, is one of the important factors for seed formation [20]. Under natural conditions, most conifer species are pollinated by wind (anemophilous pollination). Pollen can travel hundreds of miles and at thousands of feet in altitude while maintaining some viability [21,22]. A typical young reproductive *P. taeda* can shed about one kg of pollen in a two-week period. Each year, about one billion seedlings of *P. taeda* are planted in the Southeastern US, with a potential yield of ten million tons of pollen over a short rotation from this species alone. Pollen dispersal not only promotes gene flow between different populations, it also effectively increases species diversity and richness [23]. In conifer species, pollen viability and quantity are crucial for the success of pollination. Low pollen viability, high abortion rate

and unsynchronized flowering are common in conifers and thereby reduce the rate of successful zygotic fertilization [24]. Environmental conditions have a great impact on the flowering period, and pollen viability is generally reduced or lost during abnormal weather conditions [25,26]. In addition, pollen development and maturation are easily disturbed by pollutants due to industrialization [27]. Combined with the above factors, the probability of normal pollination of female cones is low in conifers, which has a direct effect on the formation and development of seed. The low viability of pollen is to some extent compensated by extraordinary pollen abundance. The synchronous release of pine pollen, for example, creates visible yellow pollen clouds, which blanket the land as a fine layer of snow. Pine pollen has been collected and distributed as a food and as a medicine for 5000 years (Tang Materia Medica 657–659 C.E.).

### 3.1.2. Temperature and Light

Low temperature affects germination by regulating the vernalization of seeds [28]. Vernalization is a process that depends on a chilling requirement to produce flowering and a good fruit crop. For example, peaches need a few hundred hours below 7 °C to satisfy the chilling requirement to break dormancy, promote flowering and have successful fruit production (https://www.greenwoodnursery.com/peach-tree-chilling-requirements accessed on 31 March 2021). The overall germination rate and the time needed for germination of seeds varies significantly under different temperature conditions [29–31]. A suitable temperature is also a precondition for flower bud formation and an important factor affecting pollen longevity and viability [32]. In addition, the number of male cones is generally more than female cones in conifers, which is not an ideal situation. Many experimental results may be caused by the different mechanisms of male and female flower buds in response to temperature, as temperature indirectly regulates the sex expression of conifer species by changing the hormone balance in flower buds [33,34]. Conifer seeds typically can be stored in the cold, e.g., at around zero degrees + or −2 °C for several months, then planted to break dormancy. Temperature signals can also regulate the activity of various enzymes and affect the metabolism in various biochemical reactions of conifer species; thus, becoming a vital participant in photosynthesis and respiration [35–37].

Light is also a crucial environmental signal and the main energy source for the plant's photosynthesis and respiration [38]. There is a high canopy density with many lateral and dead branches that affect the supply of normal light in conifers. Light intensity, spectral composition and photoperiod are important factors for conifer species growth and development, and they influence numerous physiological and biochemical reactions that cause changes in their morphology and reproductive characteristics [39]. In addition, different tree species have different demands and reaction mechanisms for light, which are affected by external environmental factors as well as their biological characters [40,41]. With the increase in forest age and the crown canopy density, the available light under the canopy will decrease, and flowering and reproduction will inevitably be affected. Therefore, an appropriate planting density, pruning and topping off can ameliorate the insufficient light of conifer species [42].

### 3.1.3. Water and Nutrient Fertilization

Water and nutrient fertilization are equally indispensable for plant seed formation and development [43]. Plant roots absorb nutrients from the soil, which is rich in minerals and organic matter to feed seed formation. Nutrient fertilization typically provides nitrogen, phosphate and potassium. Nitrogen deficiency often limits growth due to the need for substantial amounts being needed for biosynthesis of proteins and nucleic acids, while phosphorous is needed for energy metabolism and nucleic acid biosynthesis. Potassium is needed for salt balance, transport of water and nutrients. Fertilization maintains the stability of mineral circulation in the soil, which ensures the support capacity of the soil for plants and promotes the biosynthesis of proteins, amino acids and vitamins [44,45]. Plant growth and development are the result of the interaction of water and fertilizer. Water

provides a good moist environment for plant growth, which determines the activity of roots and microbes, and contributes to the construction of a good root system [46]. In the Southeastern US, with extensive plantations of southern pines, fertilization is a common practice. In this region, chronic deficiencies are found for both nitrogen and phosphorus [47]. The internal rate of return from mid-rotation fertilization of nitrogen and phosphorus was calculated to be 16%. Because most conifer species grow in mountainous areas with poor soil and harsh environmental conditions, the nutrients are typically limited in the soil [48]. Water and fertilization not only can effectively improve the soil environment, but can regulate the nutrient supply and growth, photosynthesis and other metabolic processes [49,50].

### 3.1.4. Molecular Mechanisms

The molecular mechanisms of flowering and seed production are complex, and the related genes can directly or indirectly interfere with the sex differentiation of flower, time of anthesis and seed development, resulting in the seed formation differences among individuals or species. Flower-related genes, such as *FT* (Flowering locus T) and MADS-box (*MCMI*, *AGAMOUS*, *DEFICIENCES* and *SRF* box), have been identified to play a crucial role in flowering in conifer trees.

In the study of *P. massoniana*, Chen [51,52] cloned *PmFT1* and *PmEMF2* genes by RT-PCR and RACE technology, and found that the two genes were highly expressed during the development of male and female cones, respectively, suggesting that they were involved in the development of flowers and seed formation. The *CO* gene is an important member of the regulation of sunshine length between the circadian clock and flowering time genes, which can combine light signals with circadian clock signals to regularly activate the expression of the *FT* gene; thus, inducing flowering. In *Ginkgo biloba*, the study of the effects of photoperiod on the *GbCO* gene transcription and seedling growth showed that *GbCO* activates *FT* transcription to control flowering [53,54]. The plant *LEAFY* gene encodes a class of plant-specific transcription factors, which play an important role in the transition from vegetative to reproductive growth of flowering plants. The *LEAFY* and *UFO* genes have a similar function in conifers; the *LEAFY* gene interacts with the *UFO* gene by cloning and yeast two-hybrid techniques, and it is speculated that the *UFO* gene in *Metasequoia glyptostroboides* may act as a transcriptional cofactor to regulate the *LEAFY* gene activity and, thus, participate in the regulation of the flower meristem development [55]. MADS-box genes, a class of important transcriptional regulatory factors in eukaryotes (animals, plants and fungi), play an important role in growth and development regulation and signal transduction [56,57]. There are studies that show that three MADS-box genes (*PrMADS1*, *PrMADS2* and *PrMADS3*), *LEAFY/FLORICAULA* and *NEEDLY* (*NLY*) were highly participated in the early stages of initiation and differentiation of *P. radiata* male and female cone buds, as well as vegetative buds [58]. In another study, a transcriptome data analysis showed that the enhanced transcriptional activity of MADS-box transcription factors was also closely related to the formation of early cones, in particular the process of sex reversal [59,60]. Thus, it can be also speculated that MADS-box transcription factors are a key gene family in the molecular mechanism of conifer flowering and seed production. The research on the molecular mechanism of flowering and seed production of conifers is still scarce. Therefore, it is still the mainstream direction of future research. Despite a long growth cycle and large genome compared with the broad-leaved trees, conifer species were developed and used with some traditional tree breeding methods.

### 3.2. Technical Measures of Breeding and Management

In forest tree breeding, selecting parents with good characters is the basis of tree improvement. Efficient management technology is the guarantee of a high quality and yield. Conventional methods, including artificial pollination, hormone treatment, pruning, water and fertilization coupling, can promote the flowering and reproduction of conifer species. One technical measure that greatly improves the efficient management of a

genetic improvement program is genetic fingerprinting, which provides a high level of quality control on genetic selection [61]. In addition to providing an accurate inventory, DNA markers can estimate genetic diversity, define genetic load, estimate the degree of inbreeding, identify quantitative trait loci (QTLs) and using high throughput genomic markers allow for a highly efficient genomic selection where many genomic markers associated with favorable traits can be selected simultaneously [62].

### 3.2.1. Supplementary Pollination

Plant inbreeding refers to the combination of male and female gametophytes on the same plant, or the mating between individuals with the same genotype [63]. Because of the high genetic diversity and load, close genetic relationships can lead to poor selection. Inbred lines are rare in conifer breeding programs [64,65]. The genetic characters of inbred offspring are mediocre, including growth, cone and wood traits [66]. Most conifer species are monoecious with a high possibility of inbreeding, which causes a high abortion rate of flowers and low cone yields [67]. Therefore, supplemental pollination avoids inbreeding and increases the efficiency of crosses. This method can be applied to open pollination or to controlled crosses to effectively make up for low natural pollination. The pollen with desired genes is scattered on the stigma of female flowers to create more fertilization events [66]. Understanding the details of flowering is needed to improve the sustainability and effectiveness of pollination. In the study of Douglas fir, (*Pseudotsuga menziesii*), the rate of external pollination and the inbreeding rates were 10% to 28% and 12% to 17%, respectively [68]. In addition to high inbreeding, pollen contamination from unwanted genotypes is a serious detriment to defined breeding. In Scots pine (*P. sylvestris*), a hand pollinator was used on female cones at the end of May. The amount of pollen per individual plant was 0.06~0.08 mL, and the success rate was 66–84%; whereas the success rate was 10–23% when the pollination amount per plant was 0.03–0.05 mL that was applied by a long aluminum pole duster [69]. The highest success rate of pollination was 69% using pressurized backpack sprayers to pollinate the female flowers. Therefore, methods and amounts are also critical factors affecting the success of pollination.

Supplementary pollination can introduce elite pollen into female cones to broaden the genetic base, reduce pollen contamination, increase cone yield and maximize genetic gain [70]. Understanding the flowering habits of each individual and selecting elite parents are crucial for the efficient improvement of conifer species. It is essential to select the best plants with good characters, fast growth, a high yield and allowing a robust pollen collection. Furthermore, weather condition is a key factor affecting the success rate of pollination [71]. As Additionally, the well-timed isolation of cones (method and bags) of selected mother trees has a crucial importance in the process of artificial pollination.

### 3.2.2. Topping Off

Topping off (cutting off the apical meristem) inhibits apical dominance, controls shape and height, improves light conditions and increases seed yields of trees [72]. When topping off was carried out on *P. sylvestris* for seven consecutive years, the cone yields for each individual tree increased by 95.5% and quality by 17.2% [73]. In *Pinus tabulaeformis*, the upper two-wheel branches were topped off, and the results showed an average growth rate for female cones and male cones increased by 372% and 238% [74]. Sun [75] carried out topping off on *P. koraiensis* for three years, and the best result for topping off was to cut the first wheel branch, and then the numbers of female and male cones increased by 2–4-fold. Similar results were obtained in Chen et al. [76] and Tan et al. [77] studies on *Pinus massoniana*.

Topping off can reduce the tree height of conifers, which is more convenient for cone collection and improves cone quality. The topping off treatment should be carried out in late autumn or early spring because conifer species grow slowly and have low physiological activity during those periods [74,75]. The target trees should be 10~20 m tall and have vigorous growth. For such trees, it is advisable to cut off a fifth to quarter of the tree height.

For some conifer species with a height of 20~30 m, a quarter to third of the overall height can be topped off. For particularly tall trees, the topping off height should not exceed one half of the tree height to avoid inhibition of further growth. After a topping off treatment, some new branches of conifer species will regenerate at the top in the second year, which will serve as cone setting branches, thereby increasing cone yields [74].

Adult conifer species with big crowns hardly form ideal tree shapes under natural conditions. Thus, some measures such as topping off and branch pruning should be used to control the tree shape and improve cone yields [78] (Figure 2). Many new branches are induced from new lateral meristems after topping off in the first year, then the trees create a new leader and regain apical dominance. Therefore, the topping off treatment should be implemented on lateral branches in consecutive years. The final ideal tree shape is umbrella-like and has a large canopy, low tree height, many lateral branches and new shoots [76]. When conifer species reach the mature seed production age, the number of flowers and fertile seeds increases significantly.

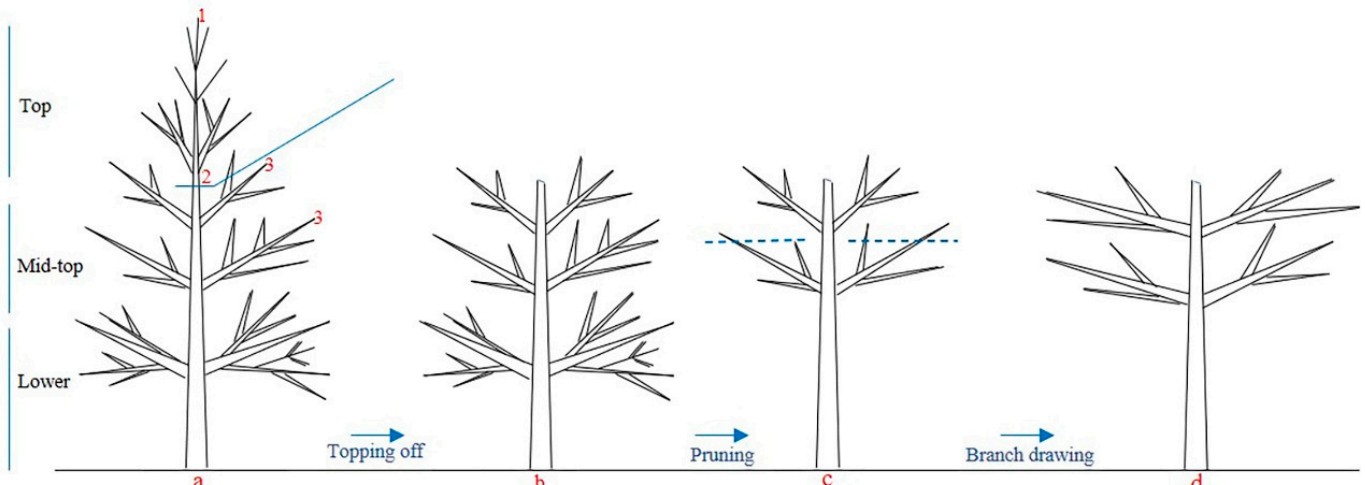

**Figure 2.** A schematic presentation of the topping off (**a**), pruning (**b**) and branch drawing (**c**) practices to improve flowering, panel (**d**) shows ideal tree shape, where in panel a 1 represents terminal buds, 2 represents topping off position on trunk and 3 represents lateral branch.

### 3.2.3. Thinning and Pruning

Thinning, a primary forestry tree management method, is the main treatment to reduce stand density. Its main function for trees is to provide light and to improve nutrient utilization efficiency [79–81], as well as carbon uptake and drought resistance [82]. Thinning allows faster growth rates, reduces root competition and combined with pruning results in longer rotation times and great improvements in wood quality. Thinning treatment of *P. koraiensis*, retaining 300 trees per hectare, with 75% intensity, increased the cone yield of each individual tree by 8~10-fold. When 750 trees per hectare were reduced by 37.5% thinning intensity, the total yield of cones increased three-fold compared to an un-thinned stand [83]. In *Pinus pinea* (stone pine), when 350 trees were retained per hectare, the cone yield of each individual tree increased by 1.48 times, while a high thinning intensity reduced the cone yield [84]. Appropriate timing and reasonable intensity are the basic requirements of thinning. Based on the evaluation of different traits such as growth, wood and cone characters, some individuals with poor performance were selected as the thinning targets [85,86]. Based on aforementioned research, the optimal thinning time was before the stage of tree sap flow or after cone shedding. For different site conditions and species characteristics, optimal thinning intensity may be different, and a poor choice of thinning methods may result in the reduction in yield [82]. In summary, conifer forests with a high density should maintain 500~700 trees per hectare after thinning in the first year, and thinning intensity should maintain 300~400 trees per hectare after 2~3 years.

Pruning is another effective measure to improve the light transmittance of trees in plantations. It refers to the removal of parts or redundant lateral branches, which can improve tree shape and increase the effect of ventilation and light transmittance [87]. Pruning may have dramatic effects on wood quality by reducing knots and promoting the growth of long straight stems. The content of trace elements in needles changes after pruning, which can improve the biomass and productivity of trees [88–90]. In *P. sylvestris* and *P. koraiensis*, the yields and quality of individual trees were significantly improved after pruning [91,92]. In another study, 3–4 main branches were kept in each round of the *P. sylvestris* crown while the remaining branches were cut off results in an increase in the number of cones of each individual tree by 14.3% [93]. Therefore, reasonable pruning plays a key role in the growth and development of conifers [94,95]. The stage of tree sap flow is the optimal time for pruning because the ability of trees to recover is strong and wound healing is fast.

### 3.2.4. Girdling and Cutting Roots

Girdling is a method involving wounding of the phloem of the trunk or main branches to regulate the distribution of nutrients in the trees so as to promote flowering and reproduction. Studies on girdling in conifers are focused on its impacts on cell activity [96], wood characteristics [97,98] and photosynthesis and respiration [99], but only a few investigate its effect on flowering and reproduction dynamics. In a study on Japanese larch (*Larix leptolepis*), girdling carried out at 1.2 m above the ground in April resulted in a significantly higher average cone yield than the control group [100].

Girdling should be carried out on trees growing in good soil conditions, with vigorous growth and no infestation/infection by pests or pathogens. The girdling time should be one month before bud differentiation. The specific method would be to girdle the bark of the trunk or main branches with a knife around the trunk with a width of 1~2 cm below the DBH (100~120 cm from the ground) and cut the bark to the xylem in depth. If the tree is very large, the girdling width can be enlarged. After 3~5 years, the number of flowers and developing seeds will improve. Girdling does damage to the tree itself, which may lead to a weakened resistance to pests and pathogens, and nutrients should be added in time to accelerate conifer growth after girdling.

Conifer species absorb nutrients from the soil by their roots. Root cutting is the process of removing excess virtual roots from the ground and limiting root growth. The study conducted on winter wheat (*Triticum aestivum*) has shown that the water consumption and water use efficiency were low after root cutting [101,102]. It also affects the photosynthesis, translocation and stress resistant of plants [103]. However, root cutting is rarely conducted in conifer species. In a study of *P. asperata*, root cutting was carried out in three seed orchards of different ages in May at about 2 m from the trunk on both sides of each tree with sharp steel plates to a soil depth of 40 cm. There was no significant difference in the number of male cones, but the number of female cones was doubled [104]. In *P. koraiensis*, a circular groove was opened at a distance of 1.7 m from the trunk to cut off roots in April, and the number of female and male flowers increased two to three-fold and one to two-fold, respectively [75]. Summing up the above, the treatment time of conifer species generally chosen before the stage of tree sap flow and an annular groove is dug around the tree 1.5~2 m away from the trunk of the tree.

### 3.2.5. Water and Fertilization Management

The number of flowers is related to the nutrient supply of the plant and the external soil environment, and some plants growing in low fertility soil produce few flowers because of nutrient deficiencies. Thus, it will be necessary to apply nutrient fertilizers and to maintain disease resistance [105,106]. Suitable water and fertilizer treatments differ according to soil conditions and biological characteristics that can effectively shorten the breeding cycle [107]. Good water and fertilizer management can not only increase production and maintain nutrient supply, but also improves photosynthesis and respiration [108,109].

The main inorganic nutrients for conifer cultivation include nitrogen (N), phosphorus (P), potassium (K), calcium (C), iron (Fe), zinc (Zn), boron (B), copper (Cu) and magnesium (Mg). The corresponding fertilization and efficacy of each element are shown in Table 1. Generally, N, P and K fertilizers are used in the management of conifer species, which play significant roles in cone development. According to different environmental conditions, the combination of many nutrient elements is needed to maintain the normal growth of plants [110]. Fertilization timing and dose are crucial for vegetative and reproductive growth of trees. In studies of conifer species, fertilizers are applied before flower bud differentiation in the spring, which can shorten flowering time of *P. taeda* clones [50]. In a study of the application of N, P, K and other fertilizers on Chinese fir (*Cunninghamia lanceolata*), the compound fertilizer (0 g N, 400 g P, 50 g K, 50 g Mg and 50 g B) increased the average number of cones in each individual plant by 0.54-fold [111]. In a study of 1.5 generations of *P. massoniana*, the application of N, P, K and micronutrient fertilizer resulted in an increase in the number of cones by 2.76-fold when 150 g N, 90 g P and 60 g micronutrients were applied to each tree. The ratio of male and female cones was more suitable for the development of seed orchards after applying 120 g N, 72 g K and 60 g of micronutrients. When the treatment involved 150 g N, 26 g P and 60 g of micronutrient being applied to each individual tree, the number of cones increased by more than three-fold [112]. In a study conducted on Fujian cypress (*F. hodginisi*), the application of 0.3 kg N, 0.4 kg P and 0.3 kg K on each individual tree increased the number of cones from each individual tree by 200% and the number of male and female cones by 129% and 338%, respectively, demonstrating that fertilization could improve the ratio of male and female cones of *F. hodginsii* and promote the flowering and reproduction of the trees [113].

**Table 1.** Fertilizer commonly used in conifer species.

| Element | Fertilizer | Efficacy | References |
|---|---|---|---|
| N | $CH_4N_2O$ | Promote flower bud development, stem and leaf growth and cone development | [114] |
| P | $Ca(H_2PO_4)_2$ | Promote plant growth and metabolism | [115] |
| K | KCl | Promote photosynthetic rate, plant resistance and cone quality | [116] |
| Ca | $Ca(NO_3)_2$ | Regulate osmotic action, enzymatic reaction and plant senescence | [112] |
| Mg | $MgSO_4$ | Promote photosynthesis and chlorophyll formation | [117] |
| B | $Na_2[B_4O_5(OH)_4]\cdot 8H_2O$ | Promote auxin operation, pollen germination and pollen tube growth | [118] |
| Cu | $CuSO_4$ | The components of enzymes | [119] |
| Zn | $ZnSO_4$ | Promote cellular respiration | [120] |

Based on the results above, we summarized a set of water and fertilization management measures for conifer species. In the stage of tree sap flow, fertilizer treatments are implemented when the soil thaws. Compound fertilizer in a ratio of N:P:K = 10:2:1 should be applied in a circular ditch 20~30 cm depth at a distance of 1.5 m away from trees. For conifer tree species with heights less than 5 m, 0~1 kg of fertilizer should be applied to each tree. The optimal dose for trees 6 to 15 m high is 1~2 kg, while the suitable dose for trees more than 15 m high is 2~5 kg. In terms of nutrients, N and P are the most important nutrients affecting formation of female cones. Besides fertilization, water is a key factor to promote the plant growth and reproduction process. In the areas of high altitude, where the soil is not suitable for storing water, irrigation is indispensable [121,122]. Dobbertin et al. [46] studied the effect of irrigation in a *P. sylvestris* forest in Switzerland by applying water quantities that doubled the long-term annual precipitation of the site. The results showed irrigation treatment increased the growth of foliage, stems and shoots, which provided basic conditions for future seed production [46]. Therefore, the conifer species planted in drought-affected areas need timely watering. However, there is a lack of reports on the related mechanisms of action between water and seed production, which will be a crucial research direction in the future. Reasonable irrigation measures and

fertilization strategies provide significant improvements in flowering and reproduction of conifers.

### 3.2.6. Phytohormone Treatments

Plant hormones are an important regulator of embryogenesis, organ development and flowering [123,124]; they can enhance stress resistance and increase yield and planting efficiency [125,126]. Phytohormones have a wide range of effects and several phytohormones affect seed formation. Auxin, (primarily indole acetic acid, IAA), gibberellins (GA), cytokinin (CTK), abscisic acid (ABA) and ethylene (ETH) are widely used in agricultural production and forest tree physiological investigations [127]. IAA, GA and CTK have the most direct activity on flowering, while ABA and ETH affect reproduction and seed production [128]. IAA is the most abundant of the auxins found in conifers, which is produced in the apical bud, then is transported down the stem and inhibits lateral bud formation. The suppression of lateral branching results in the apical dominance of the main central stem giving rise to tall straight trees able to outcompete branching trees for light and to minimize the damage to lateral branches from wind or snow. Cytokinin, known to stimulate shoot development and lateral branching are a class of molecules derived from adenine. They are synthesized in the roots and move through the xylem to leaves. In conifers, the growth and morphology of the stem and crown depend on the ratio of auxin and cytokinin.

Gibberellins (GAs) are a group of more than a hundred tetracyclic diterpenoid carboxylic acids affecting organ growth, seed dormancy, germination, flowering and senescence [129]. GAs are composed of 19 or 20 carbon atoms, and the C-19 GAs are the most active. The mutation of enzymes in the GA biosynthetic pathway are dwarfs due to the inhibition of internode growth. GAs have a key role in seed dormancy and germination in addition to metabolic effects on biosynthesis of phospholipids, nucleic acids and proteins. GAs are synthesized in the plastids and when breaking dormancy, they trigger de novo synthesis of hydrolytic enzymes in the seeds. Cytokinin and gibberellins stimulate flowering in conifers [130,131]. Ethylene is intriguing because it affects both the growth and senescence, often with contradictory effects [131]. The complexity of its effects may reside in its interaction with auxins, cytokinin, gibberellins and abscisic acid. In general, ethylene inhibits flowering and senescence, for example, in Arabidopsis and rice, but stimulates flowering in Bromeliads. In conifers, ethylene stimulates a response to wounding, to systemically acquired resistance and promotes the formation of traumatic resin ducts. Ethylene affects the branch angle (hyponasty) and, therefore, has some influence on the apical control of branching [132]. Auxin induces an ethylene response, confounding the interpretation of results because the exogenous application of auxin may induce an ethylene response.

The abundances of GA, IAA and CTK are high during bud differentiation of oriental arborvitae (*P. orientalis*), suggesting that these hormones have a significant influence on the internal mechanism of bud formation [133]. IAA and CTK are also involved in cellular proliferation and differentiation, organ formation and stem development. GA is the main regulator of the flowering pathway, and it mainly regulates growth, flower development and seed formation [134,135]. Previous studies on *G. biloba* and *P. asperata* found that the changes of hormone composition at different development stages would determine whether it would be a major or minor seed year. A high content of ZR (zeatin riboside, a cytokinin), IAA and GA are conducive to a high seed yield [136]. An imbalance of the male and female differentiation limits a high seed yield in conifer species, but the application of hormones can effectively restore the balance [137]. Table 2 summarizes studies on the application of hormones to promote flowering and reproduction in several conifer species. Among the different hormones tested, $GA_{4/7}$ was the most widely used to induce blooming in conifers [138].

**Table 2.** Studies on the effect of hormone on flowering and seed production of conifer species.

| Variety | Hormone | Dose | Effect | Method | References |
|---|---|---|---|---|---|
| | $GA_3$ | 50 mg | male bulb↑ | Ti | [139] |
| | $GA_4$ | 37 mg | female bulb↑ | Ti | |
| *Pinus koraiensis* | $GA_7$ | 37 mg | female bulb↑ | Ti | [75] |
| | $GA_3 + GA_{4/7}$ | 45 mg | male bulb↑ | Ti | |
| | 6-BA | 3 mL | female bulb↑ | S | [140] |
| *Pinus sylvestri* | | 250 mg/L | female bulb↑ | S | [141] |
| *Pinus tabuliformis* | $GA_{4/7}$ | 500 mg/L | male and female bulb↑ | S | [142] |
| *Pseudotsuga menziesii* | | 400 mg/L | male and female bulb↑ | S | [143] |
| *Pinus thunbergii Parl* | | 80 mg/L | female bulb↑ | Ti | [144] |
| *Pinus massoniana* | IAA | 250 mg/L | male and female bulb↑ | Ti | [145] |
| | BAP | 500 mg/L | male and female bulb↑ | Ti | |
| *Tsuga chinensis* | | 200 mg/L | cone↑ | Si | [146] |
| *Picea asperata Mast* | $GA_{4/7}$ | 20 mg | cone↑ | Ti | [147] |
| | | 10 mg | cone↑ | Ti | [148] |
| *Larix potaninii* | $GA_3$ | 200 mg/L | female bulb↑ | S | [149] |

↑ = increase; Ti = trunk injection; S = spraying.

In Chinese hemlock (*T. chinensis*), the cone yield was increased 2–5-fold after extensive spraying with a $GA_{4/7}$ solution [145]. The injection of $GA_{4/7}$ for two consecutive years from May to June resulted in an increase in the number of female flowers of Scots pine (*P. sylvestris*) by 63~120% and 168~282% in the first and second years, respectively [141]. In western larch (*Larix occidentalis)*, doses were graded according to the DBH so that 60 mg was injected for every 5 cm of DBH. Additionally, it was found that cone yield was increased up to seven-fold when the injection concentration was 120 mg/mL $GA_{4/7}$ [150]. In Zhao's 2007 study [142], 6-BA (6-benzylaminopurine, a cytokinin), CCC (chlormequat chloride, a GA synthesis inhibitor) and $GA_{4/7}$ were injected into plant stems, and better flowering was observed with $GA_{4/7}$ in Chinese pine (*P. tabulaeformis*). The number of female and male cones was the highest at 500 mg/L and 1000 mg/L, respectively. After a stem injection of 20 mg $GA_{4/7}$ in Chinese spruce (*P. asperata*) in June, the yield of cones for each individual increased 12-fold [147]. In another study, the number of *P. asperata* cones also increased with a hormone dose [151]. In a study of Fujian cypress (*Fokienia hodginsii*), branches were sprayed with $GA_3$, NAA (naphthaleneacetic acid, a synthetic auxin) and 2,4-D (a synthetic auxin used as an herbicide against dicots) during the flowering period. NAA had the greatest influence on the yield of male and female flowers and cones. At 200 mg/L, the increase in female flowers reached 111%, and the increase in cones was 9.27% higher than the control group [113]. In May, the number of male and female cones of Korean pine (*P. koraiensis*) was increased significantly by stem injection with $GA_3$, $GA_4$ and $GA_7$. Up to a 50 mg dose, the number of male and female cones increased with the increase in concentration [139]. Many hormones interact to regulate the various stages of flowering and seed development and germination, so hormone interactions have profound significance in promoting seed production of conifer species [143]. Receptors for endogenous hormones' action are different, which involve bud differentiation and sexual reversal by a unique signal transmission mechanism [152,153]. The balance of hormones shortened the time needed for flowering and seed formation. Under natural conditions, conifer species not only have a long breeding cycle, but also an uneven male and female gametic ratio. As the most important endogenous signaling mechanisms, phytohormones play crucial roles in the processes of growth, development, reproduction, metabolism and adaptation of conifer species [154,155].

Previous results lead to some suggestions for hormone treatments to promote flowering and seed development in conifers: (1) stem injection and spraying are the best methods for hormone treatment; (2) DBH is a useful reference for hormone dosage because the DBH growth of conifer species is directly related to the age of the tree. When DBH is below 15 cm, 30~45 mg of hormone should be injected, the concentration should be 200~350 mg/L

and spraying should continue until the branches are moist. If the DBH is 15~30 cm, hormone injection should be 45~60 mg per plant, and solution concentration should be set at 350~500 mg/L. Similarly, for every 15 cm increase in DBH, stem injection amounts should increase by 15 mg and the concentration increased to 150 mg/L. (3) Hormones should be applied before flower bud differentiation in winter and (4) GA is preferred for the promotion of flowering and seed yield. To obtain synergy in the development of male and female cones in annual cone producing conifer species, such as *Tsuga chinensis*, the application of a mixture of $GA_{4/7}$ and $GA_3$ with equal doses one month before flower bud differentiation gives the best results. The method is also suitable for some conifers with two-year growth cycles, such as *P. koraiensis*, *P. taeda* and *Pinus thunbergii*. Before snow comes, $GA_3$ should be injected to preserve pollen viability, to successfully achieve fertilization in the following year. In addition, NAA solutions can be sprayed during flowering to ensure a favorable cone-setting. CCC can be sprayed on cones twice within 7 days to ensure quality.

## 4. Conclusions and Outlook

The yield and quality of conifer cones and seeds have become a greater focus of research over the years. Many experiments about flowering and reproduction in conifers have been carried out and a substantial amount of information obtained. Based on the synthesis of available information, the present work revealed that flowering and seed production in conifers are affected by a variety of biotic and abiotic factors, That can seriously impact cone yields in plantations for commercial use. On the other hand, there are useful tools and management techniques that can be adopted in order to sustain or improve cone yields, including topping off, branch pruning, girdling, hormone application, fertilization, irrigation and supplementary pollination. Furthermore, several flowering-related genes (*FT*, Flowering locus T and MADS-box, *MCMI*, *AGAMOUS*, *DEFICIENCES* and *SRF*) that play a crucial role in flowering in conifer trees were identified. In recent years, genome selection (GS) and genetic engineering have been added to traditional approaches to conifer species breeding research. Even though conifers have very large genomes, molecular technology is being applied to the genetic improvement of economic traits for conifer species, especially in improving flowering and reproduction traits. Despite research advances, there is an urgent need to produce species-specific technical management guidelines for conifers, aiming to sustain high cone yields. This work can be considered as a first attempt of a framework to reach the goal; however, more extended and systematic research is needed.

**Author Contributions:** Conceptualization, X.-Y.Z. and Y.L.; methodology, X.L. and J.-T.W.; validation, Y.L., X.-Y.Z. and M.T.; resources, Y.L., Z.-Y.P. and M.-H.Z.; writing—original draft preparation, Y.L.; writing—review and editing, X.-Y.Z., M.T., V.L.C., H.S. and R.S.; supervision, X.-Y.Z.; project administration, X.-Y.Z.; funding acquisition, X.-Y.Z. All authors have read and agreed to the published version of the manuscript.

**Funding:** This research was funded by the Fundamental Research Funds for the Central Universities (2572020DR01, 2572020DY24) and Heilongjiang Touyan Innovation team program (Tree Genetics and Breeding Innovation Team).

**Institutional Review Board Statement:** Not applicable.

**Informed Consent Statement:** Not applicable.

**Data Availability Statement:** Data for this study can be made available with reasonable request to the author.

**Conflicts of Interest:** The authors declare that the research was conducted in the absence of any commercial of financial relationships that could be construed as a potential conflict of interest.

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
