# Peer review of "An Overview of the Practices and Management Methods for Enhancing Seed Production in Conifer Plantations for Commercial Use"

_horticulturae, doi:10.3390/horticulturae7080252_

Round 1

Reviewer 1 Report

Under the new title “Practices and Management Methods for Enhancing Seed Production in Conifer Plantations for Commercial Use”, the revised version of the manuscript is quite interesting. I acknowledge the effort by the authors to enhance the quality of their paper considering many of the suggestions mentioned in my first review report. However, there are still some important issues requiring clarification that do not allow the acceptance for publication of the manuscript with its current form. I recommend the major revision of the manuscript. Please find below my comments and suggestions.

This Review Report is based only on the revised version of the manuscript since no Report with the Responses of the authors to my comments was submitted with the revised manuscript.

A major issue that remains in the revised manuscript is that the authors characterize it as a “systematic review” paper (see line 75 in the Materials and Methods Section”. As I underlined in my previous Review Report, a Systematic Review should follow the PRISMA guidelines and this is obligatory according to the MDPI’s Journals guidelines. The authors seem that they didn’t follow the PRISMA or any other guidelines to perform a Systematic Review. They rather made a comprehensive review of selected literature, which is also useful but not as solid as a Systematic Review. Their paper can be considered as a narrative or traditional literature review and this should be clearly stated since such types of reviews are open to suggestions of bias.

In this case, I recommend that the authors change the title of the manuscript to e.g. “An Overview of the Practices and Management Methods for Enhancing Seed Production in Conifer Plantations for Commercial Use” or “Practices and Management Methods for Enhancing Seed Production in Conifer Plantations for Commercial Use-A comprehensive Overview of existing literature”. Additionally, they should mention in the abstract, materials and methods, and conclusions sections that they performed a narrative/traditional review of selected papers and that this method is subjected to possible bias. Thus the conclusions should be handled with caution.

If, on the other hand, the authors followed the PRISMA guidelines for a systematic review, this should be clearly stated in the materials and methods section by providing at least a flowchart with a description of the overall process indicating also the number of studies they identify after search, the exclusion criteria and the numbers of studies they excluded and the final number of papers they used in their review paper.

Specific other issues are also presented below:

Lines 25-27: “In this review … production” – Please rephrase

Line 28: “put forward a set of perfect technology system”. Not clear please rephrase. Reconsider the terms perfect and system.

Lines 58-59: Provide a reference. This statement is questionable considering the high density of the needles and also their high photosynthetic effectiveness. Consider also, that the effective LAI in conifers is estimated by their total leaf area i.e. both sides of the needle, whereas in broadleaved species only one side. Additionally, search about CO2 uptake from flux networks and try to justify your statement. Otherwise, remove the statement from the text.

Lines 76-77: clarify “combination”. If your review is systematic these combinations should be clear e.g. “conifer” AND “seed production” etc

Lines 79-80: Again, if the review is systematic: What were the selection criteria?, How many are these “some”, Add a flowchart. Mention the PRISMA guidelines.

Line 120: “seedlings” Is this correct? (or seeds)

Line 243: “Chen’s study. Correct to Chen et al. [76] and Tan et al. [77] studies

Line 268: “carbon sequestration” better “carbon uptake”

Lines 244-262 and 269-272: The proposed measures are quite specific and interesting. Try to add some references for justification.

Line 279: “Based on previous observations….”. Made by whom, please clarify.

Line 301: “At present, research ……”. Your references are dated from 1968 to 2008 and cannot justify the term present. Add some more recent references or delete the word “present”. Consider also removing the whole sentence.

Lines 370-376: The effect of water availability is critical. However, the statements are too general and could be a part of sub-section 3.1.3. Here try to be more generous by providing to the readers more specific and quantitative data concerning the water-tree relations and water requirements or even address the issue of irrigation water quality.

Line 433: Provide Zhao’s study and add it as a reference in the text.

Line 471 (before Pinus thunbergii) and elsewhere. Remove the hidden links

Lines 484-485: I believe that the term “complete technical system” is not appropriate. Consider the term “technical guidelines”

Figure 2. The figure is interesting. I recommend removing it to the beginning of the results section before subsection 3.1. Add also a short text with its description.

English editing is required. Check the entire manuscript.

Author Response

List of responses

Manuscript ID: horticulturae-1324563

An Overview of the Practices and Management Methods for Enhancing Seed Production in Conifer Plantations for Commercial Use

Dear Editor-in-chief and Reviewers:

Thank you for your letter and for the reviewers’ comments concerning our manuscript entitled “An Overview of the Practices and Management Methods for Enhancing Seed Production in Conifer Plantations for Commercial Use” (horticulturae-1324563). Those comments are all valuable and very helpful for revising and improving the quality our paper. We have studied all comments carefully and have made corrections which we hope meet the standard of your highly esteemed journal. For ease of tracking, we highlighted the changes with red color in the revised version. The main corrections in the paper and the responses to the reviewers’ comments are given below.

Responds to reviewer’s comments:

Reviewer #1

Under the new title “Practices and Management Methods for Enhancing Seed Production in Conifer Plantations for Commercial Use”, the revised version of the manuscript is quite interesting. I acknowledge the effort by the authors to enhance the quality of their paper considering many of the suggestions mentioned in my first review report. However, there are still some important issues requiring clarification that do not allow the acceptance for publication of the manuscript with its current form. I recommend the major revision of the manuscript. Please find below my comments and suggestions.

  • A major issue that remains in the revised manuscript is that the authors characterize it as a “systematic review” paper (see line 75 in the Materials and Methods Section”. As I underlined in my previous Review Report, a Systematic Review should follow the PRISMA guidelines and this is obligatory according to the MDPI’s Journals guidelines. The authors seem that they didn’t follow the PRISMA or any other guidelines to perform a Systematic Review. They rather made a comprehensive review of selected literature, which is also useful but not as solid as a Systematic Review. Their paper can be considered as a narrative or traditional literature review and this should be clearly stated since such types of reviews are open to suggestions of bias.

Response: The present paper is not a systematic review. Thus, we have deleted the word “systematic” from the text (see line 90).

  • In this case, I recommend that the authors change the title of the manuscript to e.g. “An Overview of the Practices and Management Methods for Enhancing Seed Production in Conifer Plantations for Commercial Use” or “Practices and Management Methods for Enhancing Seed Production in Conifer Plantations for Commercial Use-A comprehensive Overview of existing literature”. Additionally, they should mention in the abstract, materials and methods, and conclusions sections that they performed a narrative/traditional review of selected papers and that this method is subjected to possible bias. Thus, the conclusions should be handled with caution.

Response: We have changed the title of the manuscript to “An Overview of the Practices and Management Methods for Enhancing Seed Production in Conifer Plantations for Commercial Use”, and it was mentioned in the abstract, materials and methods, and conclusions sections (see the new title).

  • If, on the other hand, the authors followed the PRISMA guidelines for a systematic review, this should be clearly stated in the materials and methods section by providing at least a flowchart with a description of the overall process indicating also the number of studies they identify after search, the exclusion criteria and the numbers of studies they excluded and the final number of papers they used in their review paper.

Response: The present paper is not a systematic review. We have deleted the word “systematic” from the text.

  • Lines 25-27: “In this review … production” – Please rephrase

Response: The sentence has been modified (see line 26-28).

  • Line 28: “put forward a set of perfect technology system”. Not clear please rephrase. Reconsider the terms perfect and system.

Response: The sentence has been modified (see line 29-30).

  • Lines 58-59: Provide a reference. This statement is questionable considering the high density of the needles and also their high photosynthetic effectiveness. Consider also, that the effective LAI in conifers is estimated by their total leaf area i.e., both sides of the needle, whereas in broadleaved species only one side. Additionally, search about CO2 uptake from flux networks and try to justify your statement. Otherwise, remove the statement from the text.

Response: The sentence has been deleted from the text.

  • Lines 76-77: clarify “combination”. If your review is systematic these combinations should be clear e.g., “conifer” AND “seed production” etc.

Response: We have added some key words to this paragraph (see line 94-95).

  • Lines 79-80: Again, if the review is systematic: What were the selection criteria? How many are these “some”, Add a flowchart. Mention the PRISMA guidelines.

Response: The present paper is not a systematic review. We have deleted the word “systematic” from the text.

  • Line 120: “seedlings” Is this correct? (Or seeds)

Response: The word “seedlings” has been replaced by “seeds” (see line 185).

  • Line 243: “Chen’s study. Correct to Chen et al. [76] and Tan et al. [77] studies

Response: The information has been corrected (see line 311).

  • Line 268: “carbon sequestration” better “carbon uptake”

Response: The phrase “carbon sequestration” has been replaced by “carbon uptake” (see line 341).

  • Lines 244-262 and 269-272: The proposed measures are quite specific and interesting. Try to add some references for justification.

Response: Thanks for your review. We have added a few literatures for justification (see line 322-340).

  • Line 279: “Based on previous observations….”. Made by whom, please clarify.

Response: The sentences have been refined and clarified (see line 352).

  • Line 301: “At present, research ……”. Your references are dated from 1968 to 2008 and cannot justify the term present. Add some more recent references or delete the word “present”. Consider also removing the whole sentence.

Response: The word “present” have been deleted.

  • Lines 370-376: The effect of water availability is critical. However, the statements are too general and could be a part of sub-section 3.1.3. Here try to be more generous by providing to the readers more specific and quantitative data concerning the water-tree relations and water requirements or even address the issue of irrigation water quality.

Response: We have supplemented and perfected the content in detail (see line 510-517).

  • Line 433: Provide Zhao’s study and add it as a reference in the text.

Response: The reference has been added in the text (see line 678).

  • Line 471 (before Pinus thunbergii) and elsewhere. Remove the hidden links

Response: The hidden links have been removed (see line 766).

  • Lines 484-485: I believe that the term “complete technical system” is not appropriate. Consider the term “technical guidelines”

Response: The “complete technical system” has been replaced by “technical guidelines” (see line 779-780).

  • Figure 2. The figure is interesting. I recommend removing it to the beginning of the results section before subsection 3.1. Add also a short text with its description.

Response: Figure 2. has been removed to the beginning of the results section before subsection 3.1, and we add a short description.

  • English editing is required. Check the entire manuscript.

Response: We have changed English language and style.

Reviewer 2 Report

The draft presents a nice and very clear review of the methods for increasing seed production in conifer’s plantations.  Usually this technics have been used in seed orchards for seed supply oriented to afforestation in tree breeding programs but can be used for any other use of the seed (food, medical, etc.).

The technics are presented one by one in a clear way quite useful for teaching and/or updating of professionals. The combination of classic methods (i.e. girdling or topping) with the new genetic advances (transcription factor genes) are showed in a comprehensive full picture.

Only two very minor suggestions can offer to improve the reading of the manuscript:

Line  90: change “a” by “one”

Line 162: write a short description of the acronyms, in brackets for example, of the FT and MADS-box genes (“Flowering locus T” and “Transcription factor genes”)

Author Response

List of responses

Manuscript ID: horticulturae-1324563

An Overview of the Practices and Management Methods for Enhancing Seed Production in Conifer Plantations for Commercial Use

Dear Editor-in-chief and Reviewers:

Thank you for your letter and for the reviewers’ comments concerning our manuscript entitled “An Overview of the Practices and Management Methods for Enhancing Seed Production in Conifer Plantations for Commercial Use” (horticulturae-1324563). Those comments are all valuable and very helpful for revising and improving the quality our paper. We have studied all comments carefully and have made corrections which we hope meet the standard of your highly esteemed journal. For ease of tracking, we highlighted the changes with red color in the revised version. The main corrections in the paper and the responses to the reviewers’ comments are given below.

Responds to reviewer’s comments:

Reviewer #2

The draft presents a nice and very clear review of the methods for increasing seed production in conifer’s plantations. Usually, these technics have been used in seed orchards for seed supply oriented to afforestation in tree breeding programs but can be used for any other use of the seed (food, medical, etc.).

The technics are presented one by one in a clear way quite useful for teaching and/or updating of professionals. The combination of classic methods (i.e., girdling or topping) with the new genetic advances (transcription factor genes) are showed in a comprehensive full picture.

Only two very minor suggestions can offer to improve the reading of the manuscript:

  • Line 90: change “a” by “one”

Response: The word “a” has been replaced by “one” (see line 145).

  • Line 162: write a short description of the acronyms, in brackets for example, of the FT and MADS-box genes (“Flowering locus T” and “Transcription factor genes”)

Response: We have made a short description of the acronyms (see line 229-230).

Reviewer 3 Report

Dear authors,

thank you for presentented paper. Review would fit more as technical report or chapter in a book. 

It would be nice to present more suggestions of possible production cycles especialy based on results of field trials.

Author Response

List of responses

Manuscript ID: horticulturae-1324563

An Overview of the Practices and Management Methods for Enhancing Seed Production in Conifer Plantations for Commercial Use

Dear Editor-in-chief and Reviewers:

Thank you for your letter and for the reviewers’ comments concerning our manuscript entitled “An Overview of the Practices and Management Methods for Enhancing Seed Production in Conifer Plantations for Commercial Use” (horticulturae-1324563). Those comments are all valuable and very helpful for revising and improving the quality our paper. We have studied all comments carefully and have made corrections which we hope meet the standard of your highly esteemed journal. For ease of tracking, we highlighted the changes with red color in the revised version. The main corrections in the paper and the responses to the reviewers’ comments are given below.

Responds to reviewer’s comments:

Reviewer #3

  • Thank you for presentenced paper. Review would fit more as technical report or chapter in a book. It would be nice to present more suggestions of possible production cycles especially based on results of field trials.

Response: Dear reviewer, thank you for your suggestions. We are doing some field trials related to water-fertilizer coupling, hormone treatment and tree management, which will provide the practical information for conifer species in the future.

Round 2

Reviewer 1 Report

The revised version of the manuscript is significantly improved. In my opinion, it can be considered for publication after minor revisions. More specifically see my comments below.

Language editing is necessary. In many parts of the revised text grammar errors should be corrected.

Lines 24-25: I prefer the text before the correction. In any case, it needs to be rephrased.

Lines 27-33: Consider rephrasing the sentences “In this review, to improve the problems of low flowering and seed yield, we present a traditional literature review of recent research progress in conifer flowering and seed production. Discussing many factors that affect flowering and seed production of conifer species, we finally put forward a comprehensive practices and management methods for enhancing seed production in conifer plantations for commercial use, which may provide more distinct insight for future research on conifer species” to

“In the present work, we reviewed selected literature and provide a comprehensive overview on the most influential factors and also on the methods and techniques that can be adopted, in order to improve flowering and seed production in conifers species. The results of this study can be useful for forest managers and also for enhancing seed yields in conifers plantations for commercial use.”

Lines 41-43: Consider rephrasing the sentence “Conifers include about 613 species, mainly included Pinaceae, Taxodiaceae and Cupressaceae that are widely distributed over the world [2, 3]” to

“Conifers include about 613 species, mainly included in the Pinaceae, Taxodiaceae and Cupressaceae families, and are widely distributed over the world [2, 3]”

Lines 43-44: Consider rephrasing the sentence “Conifers are some of the first living things on land and have been around for about three hundred million years.” to

“They appeared on earth’s surface about three hundred million years ago.”

Also, add a reference.

Lines 80-83: Consider rephrasing the sentence “A comprehensive overview literature review of selected literature was conducted in this review, searching the platforms Web of Science, Google scholar and Endnote literature management software for a combination of the following keywords: …” to

“For the current study, we reviewed articles indexed in the databases Web of Science and Google Scholar. At the first stage, relevant studies were identified by using combinations of the following keywords: …. At the final stage, the most appropriate articles were selected, to perform a solid overview of a) the factors affecting conifers’ flowering and seed production and b) the techniques and management practices that are can be adopted to enhance seed production.”

Also, consider that Endnote is not a platform to search papers. As you mention is a literature management tool.

Lines 93: “the complete” remove.

Line 94: “conifer plantations and commercial use” change to “conifer plantations for commercial use”.

Line 95: Add a paragraph for the description of Fig. 1, providing an overview of the following chapters. You can start by stating that flowering and seed yields in conifers are affected by a variety of factors (pollen, temperature, light, water availability, nutrients, etc) and that for improving cone yields a number of management techniques have been tested including toping, pruning, fertilization, hormone regulation, supplementary pollination, etc, that are presented in the graphical abstract of figure 1.

Line 98: Provide an explanation for the “5~10a”

Line 108: Choose between “in a two-week period” or “in two weeks”

Lines 323-326: Consider rephrasing the text “There were some researches on girdling in conifers focuses on cell activity [96], wood characteristics [97, 98], photosynthesis and respiration [99]. However, there are few studies on girdling related to flowering and reproduction.” to

“Studies on girdling in conifers are focused on its impacts on cell activity [96], wood characteristics [97, 98], and photosynthesis and respiration [99], but only a few investigate its effect on flowering and reproduction dynamics”

Lines 397-398: Consider rephrasing the text “In the P. sylvestris study, trees were irrigated throughout the vegetation period during rainless nights, which doubled the long-term annual precipitation at the site” to

Dobbertin et al. [46] studied the effect of irrigation in a P. sylvestris forest in Switzerland, by applying water quantities that doubled the long-term annual precipitation of the site.

Lines 509-524: Consider rephrasing the text, in order to highlight the findings of your work. You can add something of the following:

Based on the results of the present work, flowering and seed production in conifers are affected by a variety of biotic and abiotic factors, that can seriously impact cone yields in plantations for commercial use. On the other hand, there are useful tools and management techniques that can be adopted in order to sustain or improve cone yields, including …………….

The above impose an urgent need to produce specific technical management guidelines for conifer plantations and forests, aiming to sustain high cone yields. This work can be considered as a first attempt of a framework to reach the goal, however more extended and systematic research is needed.

Author Response

List of responses

Manuscript ID: horticulturae-1324563

An Overview of the Practices and Management Methods for Enhancing Seed Production in Conifer Plantations for Commercial Use

Dear Editor-in-chief and Reviewers:

Thanks for your letter again and for the reviewers’ comments concerning our manuscript entitled “An Overview of the Practices and Management Methods for Enhancing Seed Production in Conifer Plantations for Commercial Use” (horticulturae-1324563). Those comments are all valuable and very helpful for revising and improving the quality our paper. We have studied all comments carefully and have made corrections which we hope meet the standard of your highly esteemed journal. For ease of tracking, we highlighted the changes with red color in the revised version. The main corrections in the paper and the responses to the reviewers’ comments are given below.

Responds to reviewer’s comments:

Reviewer #1

The revised version of the manuscript is significantly improved. In my opinion, it can be considered for publication after minor revisions. More specifically see my comments below.

Response: Thank you for the compliments!

Language editing is necessary. In many parts of the revised text grammar errors should be corrected.

Response: The language is thoroughly checked by native English-speaking colleague.

  • Lines 24-25: I prefer the text before the correction. In any case, it needs to be rephrased.

Response: We rephrased the sentence as per the suggestion (see line 24-28).

  • Consider rephrasing the sentences “In this review, to improve the problems of low flowering and seed yield, we present a traditional literature review of recent research progress in conifer flowering and seed production. Discussing many factors that affect flowering and seed production of conifer species, we finally put forward a comprehensive practices and management methods for enhancing seed production in conifer plantations for commercial use, which may provide more distinct insight for future research on conifer species” to

“In the present work, we reviewed selected literature and provide a comprehensive overview on the most influential factors and also on the methods and techniques that can be adopted, in order to improve flowering and seed production in conifers species. The results of this study can be useful for forest managers and also for enhancing seed yields in conifers plantations for commercial use.”

Response: We thank the reviewer for the constructive suggestion. Changes are made as suggested (see line 26-29).

  • Lines 41-43: Consider rephrasing the sentence “Conifers include about 613 species, mainly included Pinaceae, Taxodiaceae and Cupressaceae that are widely distributed over the world [2, 3]” to

“Conifers include about 613 species, mainly included in the Pinaceae, Taxodiaceae and Cupressaceae families, and are widely distributed over the world [2, 3]”

Response: We thank the reviewer for the constructive suggestion. Changes are made as suggested (see line 44-45).

  • Lines 43-44: Consider rephrasing the sentence “Conifers are some of the first living things on land and have been around for about three hundred million years.” to

“They appeared on earth’s surface about three hundred million years ago.”

Also, add a reference.

Response: We thank the reviewer for the constructive suggestion. Changes are made as suggested (see line 45-46).

  • Lines 80-83: Consider rephrasing the sentence “A comprehensive overview literature review of selected literature was conducted in this review, searching the platforms Web of Science, Google scholar and Endnote literature management software for a combination of the following keywords: …” to

“For the current study, we reviewed articles indexed in the databases Web of Science and Google Scholar. At the first stage, relevant studies were identified by using combinations of the following keywords: …. At the final stage, the most appropriate articles were selected, to perform a solid overview of a) the factors affecting conifers’ flowering and seed production and b) the techniques and management practices that are can be adopted to enhance seed production.”

Also, consider that Endnote is not a platform to search papers. As you mention is a literature management tool.

Response: We thank the reviewer for the constructive suggestion. Changes are made as suggested (see L82-89). We excluded the Endnote in the text as we completely agree with the reviewer that it is rather a literature management tool.

  • Lines 93: “the complete” remove.

Response: We have changed made as suggested (see line 92).

  • Line 94: “conifer plantations and commercial use” change to “conifer plantations for commercial use”.

Response: We have changed made as suggested (see line 93).

  • Line 95: Add a paragraph for the description of Fig. 1, providing an overview of the following chapters. You can start by stating that flowering and seed yields in conifers are affected by a variety of factors (pollen, temperature, light, water availability, nutrients, etc.) and that for improving cone yields a number of management techniques have been tested including toping, pruning, fertilization, hormone regulation, supplementary pollination, etc., that are presented in the graphical abstract of figure 1.

Response: We thank the reviewer for the constructive suggestion. Changes are made as suggested (see line 95-101).

  • Line 98: Provide an explanation for the “5~10a”

Response: we rewritten this as 5~10 years (see line 98).

  • Line 108: Choose between “” or “in two weeks”

Response: Thank you for suggested. We opted for a two-week period (see line 108).

  • Lines 323-326: Consider rephrasing the text “There were some researches on girdling in conifers focuses on cell activity [96], wood characteristics [97, 98], photosynthesis and respiration [99]. However, there are few studies on girdling related to flowering and reproduction.” to

“Studies on girdling in conifers are focused on its impacts on cell activity [96], wood characteristics [97, 98], and photosynthesis and respiration [99], but only a few investigate its effect on flowering and reproduction dynamics”

Response: We thank the reviewer for the constructive suggestion. Changes are made as suggested (see line 324-326).

  • Lines 397-398: Consider rephrasing the text “In the P. sylvestris study, trees were irrigated throughout the vegetation period during rainless nights, which doubled the long-term annual precipitation at the site” to

Dobbertin et al. [46] studied the effect of irrigation in a P. sylvestris forest in Switzerland, by applying water quantities that doubled the long-term annual precipitation of the site.

Response: We thank the reviewer for the constructive suggestion. Changes are made as suggested (see line 395-397).

  • Lines 509-524: Consider rephrasing the text, in order to highlight the findings of your work. You can add something of the following:

Based on the results of the present work, flowering and seed production in conifers are affected by a variety of biotic and abiotic factors, that can seriously impact cone yields in plantations for commercial use. On the other hand, there are useful tools and management techniques that can be adopted in order to sustain or improve cone yields, including …………….

The above impose an urgent need to produce specific technical management guidelines for conifer plantations and forests, aiming to sustain high cone yields. This work can be considered as a first attempt of a framework to reach the goal, however more extended and systematic research is needed.

Response: We thank the reviewer for the constructive suggestion. Changes are made as suggested (see line 505-523).

Reviewer 3 Report

Dear authors,

please check following parts/ questions:

Material and methods - Is the selection of key word based on other previous review/studies? Why you selected exactely  these words?

Row

87 - "some papers" how many? What is the meaning "some"?

94 - please check formulation of sentence

248 - also well-timed isolation of cones (method and bags) of selected mother trees has an crucial importance in process of artificial pollination (pollen storage also/ methods and duration). Controlling of cones development and well-timed removing of isolation bags too.

Fig2. - schema/cycle/process to ...

Some techniques to synchronize flowering?

Thank you

Author Response

List of responses

Manuscript ID: horticulturae-1324563

An Overview of the Practices and Management Methods for Enhancing Seed Production in Conifer Plantations for Commercial Use

Dear Editor-in-chief and Reviewers:

Thanks for your letter again and for the reviewers’ comments concerning our manuscript entitled “An Overview of the Practices and Management Methods for Enhancing Seed Production in Conifer Plantations for Commercial Use” (horticulturae-1324563). Those comments are all valuable and very helpful for revising and improving the quality our paper. We have studied all comments carefully and have made corrections which we hope meet the standard of your highly esteemed journal. For ease of tracking, we highlighted the changes with red color in the revised version. The main corrections in the paper and the responses to the reviewers’ comments are given below.

Responds to reviewer’s comments:

Reviewer #3

  • Material and methods - Is the selection of key word based on other previous review/studies? Why you selected exactly these words?

Response: The keywords are selected based on the information we would like to retrieve from the database. We believe that these keywords are appropriate and satisfactory to search the information we are interested in.

  • 87 - "some papers" how many? What is the meaning "some"?

Response: The sentence is rephrased according to the suggestion by Reviewer #1 (see line 87-89).

  • 94 - please check formulation of sentence

Response: The sentences is rephrased (see line 91-93).

  • 248 - also well-timed isolation of cones (method and bags) of selected mother trees has a crucial importance in process of artificial pollination (pollen storage also/ methods and duration). Controlling of cones development and well-timed removing of isolation bags too.

Response: We added the suggested text in the revised version (see line 251-253).

  • - schema/cycle/process to ...

Some techniques to synchronize flowering?

Response: The figure caption is rephrased as per the suggestion (see line 285-287).

This manuscript is a resubmission of an earlier submission. The following is a list of the peer review reports and author responses from that submission.

Round 1

Reviewer 1 Report

The submitted paper titled “Flowering and seed production in conifer species: a systematic review” attempts to present the advances in flowering and seed production in some conifers, aiming (according to the authors) to increase the numbers of flowers and seeds.

The topic is interesting and within the scope of the journal, however, conifers include an extremely large taxa with a great number of families with different characteristics, whereas their flowering and seed production alter also according to the genetic and epigenetic characteristics in different parts of the globe.

One of my concerns for this paper is that the authors characterized it as “a systematic review”. According to the journal’s “Instructions to authors” presented in https://www.mdpi.com/journal/horticulturae/instructions, systematic reviews should “use the same structure as research articles and ensure they conform to the PRISMA guidelines” (please refer to http://prisma-statement.org/). The authors do not appear to have followed these guidelines and they do not mention any other methodology adopted in order to make a systematic review.

Another important issue is the references used by the authors to justify their claims and statements (e.g. lines 160-162 etc). Most of them refer to agricultural crops as tomato (ref. 71), rice (ref. 72), wheat (ref.73m 93), maize, strawberries (ref. 202), non-tree species as A. thaliana (Brassicaceae family, ref. 89, 90, 91, 99, 100), tabacco (ref. 92), or non-conifer trees as poplar (ref 94), Ginco biloba (ref. 97, 98), apples (ref. 201), peaches (ref. 203), etc. These references should be at a minimum and only when they are necessary for comparison between species.

The introduction is very general and not specific to the flowering and seed production of conifers. I suggest being expanded by incorporating additional findings for flowering and seed production for conifer species in different parts of the world. Also at the end of the introduction, the authors should present the scope of their study and the very specific research questions they try to answer with their review paper. Try to make these questions clear and specific.

Add a section as “Materials and Methods” to present the methods used for the search of the scientific papers. Present how the relevance of the selected paper was evaluated. According to the journal’s “instructions to authors,” you should follow a specific methodology and you should describe all steps when writing a review according to PRISMA guidelines. Refer to the databases used, the number of papers selected, how many of them were excluded and with what criteria, what were the keywords used, what was the final number of papers you included in your work, etc.

Sections 2 and 3 provide too general information and most of it has to do with factors affecting in general plants physiology and growth (not only for conifers). For example, in subsection 3.3 (Water and nutrient fertilization” extended and too general information is provided for the plants' physiology and growth processes with many references in agricultural crops, without indicating a relationship with flowering and seed production even for these species. Also, in lines 198-217 an extended analysis is provided for not conifer species. I suggest diminishing the length of these sections to a minimum and concentrate on conifers’ flowering and seed production.

Sections 4 and 5 are, in my opinion, the main and most interesting parts of the paper, presenting techniques and management methods to increase seed production in conifers. I would only add some references indicating that there is a need to increase seed production in specific natural conifer forests or when managing conifer plantations for commercial use. Based on these sections, the paper should be extensively revised and in my opinion, it should be resubmitted under a different title as “Measures and techniques for increasing seed production in conifers” or “Practices and management methods for enhancing/improving seed production in conifer plantations for commercial use” etc

The conclusion section should be specific and in line with the aim of your work, concentrated in conifers. Any references for agricultural crops should be removed.

The number of references is high (206 refs), which is common for review papers. However, as mentioned above, a great number of them are not relevant to the subject of the paper. Try to exclude all unnecessary ones.

Based on the above-mentioned comments and suggestions, I recommend the rejection of the manuscript for publication in its current form. The authors should resubmit their paper under a different title, presenting also a solid methodological approach and a concise analysis based on conifers’ seed production.